# Development of a Semi-Analytical Solution for Simulating the Migration of Parent and Daughter Contaminants from Multiple Contaminant Sources, Considering Rate-Limited Sorption Effects

Thu-Uyen Nguyen [1], Yi-Hsien Chen [1], Heejun Suk [2], Ching-Ping Liang [3] and Jui-Sheng Chen [1,4,*]

1 Graduate Institute of Applied Geology, National Central University, Taoyuan City 320317, Taiwan; 110684603@cc.ncu.edu.tw (T.-U.N.); 109624005@cc.ncu.edu.tw (Y.-H.C.)
2 Korea Institute of Geoscience and Mineral Resources, Daejeon 34132, Republic of Korea; sxh60@kigam.re.kr
3 Department of Nursing, Fooyin University, Kaohsiung City 83101, Taiwan; sc048@fy.edu.tw
4 Center for Advanced Model Research Development and Applications, National Central University, Taoyuan City 320317, Taiwan
* Correspondence: jschen@geo.ncu.edu.tw

## Abstract

Most existing multispecies transport analytical models primarily focus on inlet boundary sources, limiting their applicability in real-world contaminated sites where contaminants often arise from multiple internal sources. This study presents a novel semi-analytical model for simulating multispecies contaminant transport driven by multiple time-dependent internal sources. The model incorporates key transport mechanisms, including advection, dispersion, rate-limited sorption, and first-order degradation. In particular, the inclusion of rate-limited sorption addresses limitations in traditional equilibrium-based models, which often underestimate pollutant concentrations for degradable species. The derivation of this semi-analytical model utilizes the Laplace transform, finite cosine Fourier transform, generalized integral transform, and a sequence of inverse transformations. Results indicate that the concentrations of contaminants and their degradation products are highly sensitive to the variations in time-dependent sources. The model's most significant contribution lies in its capability to simulate the contaminant transport from multiple internal pollution sources at a contaminated site under the influence of rate-limited sorption. By enabling the representation of multiple time-varying sources, this model fills a critical gap in analytical approaches and provides a necessary tool for accurately assessing contaminant transport in complex, realistic pollution scenarios.

**Keywords:** multiple contaminant sources; multi-species transport; rate-limited sorption; semi-analytical solution

## 1. Introduction

Improperly managed waste from industrial processes, agricultural practices, and other human activities can lead to severe environmental consequences. These pollutants may infiltrate the soil and percolate into groundwater systems through surface pathways, triggering regional contamination events that pose serious risks to drinking water safety. Understanding the transport behavior of contaminants in groundwater is therefore essential for effective pollution control and for selecting remediation strategies suited to the specific conditions of each site.

To better understand the movement and transformation of contaminants in groundwater systems, various mathematical models describing solute transport have been developed

over the past decades. Both analytical and numerical solution models have been proven effective in simulating contaminant transport in groundwater systems. Although analytical models rely on simplifying assumptions, they are easier to implement when data availability is limited. The selection between analytical and numerical approaches depends on the modeling objectives, data availability, site complexity, and budgetary constraints. When extensive field data are available, three-dimensional numerical models provide powerful simulation capabilities. However, for most practical applications, data are often scarce. In such cases, analytical models offer a valuable means for obtaining preliminary estimates of contaminant plume migration. For example, the BIOCHLOR model developed by Aziz et al. (2000) [1], a widely used and publicly accessible analytical tool, simulates three-dimensional multispecies transport and is particularly effective in modeling natural attenuation at chlorinated solvent-contaminated sites. McGuire et al. (2004) [2] reported that among 45 such sites, 60% employed mathematical models, with BIOCHLOR being the most commonly used. Accordingly, many analytical models have been developed over the past decades to simulate one-, two-, and three-dimensional contaminant transport based on advection-dispersion equations (ADEs). Numerous models focusing on single-species transport processes have been documented in the literature [3–9].

The models developed in the aforementioned studies primarily focus on simulating the transport behavior of single-species solutes. However, certain pollutants—such as chlorinated solvents, radionuclides, and nitrogen compounds—often degrade under favorable environmental conditions, producing secondary compounds known as daughter products. These transformations typically follow a sequential first-order decay process, governed by first-order reaction kinetics, resulting in dynamic changes in the concentrations of both parent and daughter species. Since each compound in the degradation chain is interdependent, the concentration of a daughter product may increase or decrease depending on the reaction rate constant of its parent compound [10]. As a result, single-species solute transport models are inherently incapable of capturing the mass transformation processes associated with degradable contaminants. In contrast, multispecies transport models are essential for accurately representing the mass conversion from parent to daughter species in reactive transport systems.

Over the past decades, numerous models have been developed to simulate the transport behavior of multiple contaminants in groundwater systems and have been widely applied in environmental studies [11–18]. Most existing analytical models assume linear equilibrium sorption, where the sorption rate between dissolved and solid phases is significantly faster than the advective transport of pollutants through porous media. However, this assumption can lead to inaccurate predictions of contaminant concentrations—particularly under conditions of rapid groundwater flow—resulting in significant underestimation of pollutant levels and associated health risks [19–21]. Such models fail to accurately capture transport dynamics at sites where sorption does not reach equilibrium instantaneously [22]. Although developing analytical solutions that incorporate rate-limited (nonequilibrium) sorption is more complex, such models provide more realistic representations of contaminant behavior and are better suited for addressing practical field conditions.

Most existing analytical models treat the contaminant source as an inlet boundary condition [23], which often oversimplifies real-world scenarios. In practice, many contaminated sites involve multiple distributed sources within the domain, such as leakage points from pipelines, tanks, or subsurface waste deposits. To realistically represent such complex systems, it is essential to formulate the advection–dispersion equation with source/sink terms that explicitly account for internal and multiple pollution sources. These source/sink terms serve as crucial components in capturing the spatial and temporal variability of contaminant inputs across the domain. Despite their importance, analytical solutions

incorporating multiple internal sources remain scarce due to the increased mathematical complexity compared to traditional boundary-driven models. The development of such models is thus urgently needed to improve the applicability and accuracy of analytical solutions in real-world groundwater contamination assessments.

Chen et al. [24] developed analytical models to simulate the transport of a single contaminant with arbitrary one-, two-, and three-dimensional source geometries, enabling the placement of pollution sources at any location within the domain. However, these models are limited to a single internal source. In real-world scenarios, multiple pollution sources are frequently encountered, such as leaks from sewage pipes, leachate from landfill liners, and oil spills from storage tanks—often resulting from structural defects or failures in pipelines and containment systems. To address this complexity, Ding et al. [25] proposed a two-dimensional analytical solution for single-species solute transport that incorporates multiple time-varying internal point sources within a finite domain. While their model accounts for internal source injection, it lacks the capacity to simulate the formation and transport of degradation products.

This study introduces a novel semi-analytical model for simulating the transport of multi-species contaminants in groundwater systems, explicitly accounting for rate-limited sorption and multiple time-dependent internal sources distributed arbitrarily within the domain. Unlike conventional analytical models that typically assume boundary-driven inputs, the proposed model incorporates source/sink terms into the coupled ADEs to better reflect real-world scenarios involving spatially distributed contamination events. To derive the solution efficiently, the governing equations are transformed into a system of algebraic equations using the Laplace transform in the time domain and integral transforms in the spatial dimensions. This modeling framework enables more realistic simulation of contaminant plume migration and transformation, particularly for degradable pollutants and their daughter products in complex subsurface environments.

## 2. Mathematical Model

This study presents a semi-analytical two-dimensional model for evaluating the transport of contaminants and their degradation products in groundwater, incorporating first-order reaction kinetics and the effects of rate-limited sorption. The model accounts for multiple contaminant sources distributed arbitrarily within the domain. Transport processes include advection, hydrodynamic dispersion, first-order degradation, and reversible sorption kinetics. The primary contaminant is subject to rate-limited sorption, modeled using first-order reversible kinetics, while its degradation products follow sequential first-order decay reactions. These processes are critical for accurately capturing the transformation and migration of degradable contaminants in subsurface environments. As illustrated in Figure 1, the model describes multispecies contaminant transport in a homogeneous aquifer with unidirectional groundwater flow along the x-axis. Contaminant sources are represented as horizontally oriented rectangles positioned at arbitrary locations. Dispersion occurs in both the longitudinal (x) and transverse (y) directions.

The governing equations characterize the coupled transport of the parent contaminant and its daughter products, incorporating multiple source terms and first-order reversible sorption in two spatial dimensions as

$$D_L \frac{\partial^2 C_i(x,y,t)}{\partial x^2} - v \frac{\partial C_i(x,y,t)}{\partial x} + D_T \frac{\partial^2 C_i(x,y,t)}{\partial y^2} - k_i C_i(x,y,t) + k_{i-1} C_{i-1}(x,y,t) + \sum_{m=1}^{NS} \frac{M_i^m}{\phi} p_{x,i}^m(x) p_{y,i}^m(y) q_i^m(t) -$$
$$\frac{\beta_i}{\phi} \left( C_i(x,y,t) - \frac{S_i(x,y,t)}{K_{di}} \right) = \frac{\partial C_i(x,y,t)}{\partial t} \qquad i = 1, 2, \ldots, N \tag{1}$$

$$\rho_b \frac{\partial S_i(x,y,t)}{\partial t} = \beta_i \left( C_i(x,y,t) - \frac{S_i(x,y,t)}{K_{di}} \right) \qquad i = 1, 2, \ldots, N \tag{2}$$

In the governing equation, $C_i(x,y,t)$ denotes the concentration of the *ith* species in the aqueous phase (M/L$^3$). $S_i(x,y,t)$ is the concentration of the *ith* species adsorbed to the solid phase (M/M) and *t* denotes time (T). $D_L$ and $D_T$ are the longitudinal and lateral dispersion coefficients (L$^2$/T), while $v$ is the average pore-water velocity (L/T). $k_i$ is the decay/degradation rate constant of the *ith* species (T$^{-1}$). The parameter β represents the kinetic sorption rate constant (T$^{-1}$). Porosity is denoted by θ (-), and $\rho_b$ is the bulk density of the geological material (M/L$^3$). The distribution of the ith species between the aqueous and solid phase is characterized by the partition coefficient $K_{di}$ (L$^3$/M). The source/sink term is expressed as $\sum_{m=1}^{NS} \frac{M_i^i}{\phi} p_{x,i}^m(x) p_{y,i}^m(y) q_i^m(t)$ , NS is the total number of contaminant sources, $M_i^i$ is the mass of species *i* introduced at source and $p_{x,i}^m(x)$ and $p_{y,i}^m(y)$ are the spatial distribution functions for the x and y directions, respectively. These spatial functions are typically represented by unit step functions. $q_i^m(t)$ defines the temporal variation in species *i* at source *m*. In this study, two types of time-dependent contaminant sources are considered: (1) instantaneous Dirac delta function and (2) finite pulse function, as summarized in Table 1.

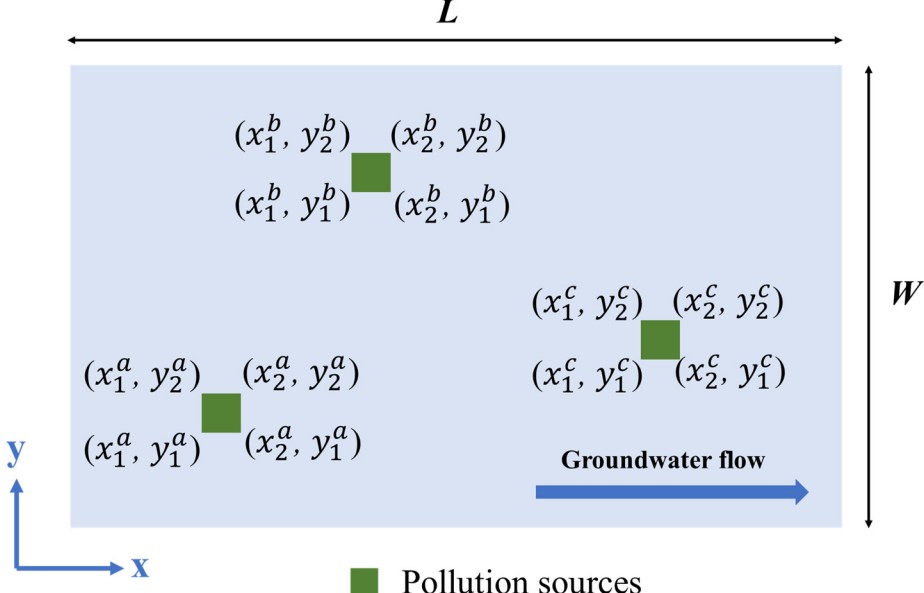

**Figure 1.** Schematic of the model domain for a multi-source system. Green squares (sources a–c) denote source zones; for each source, the footprint is defined by the corner coordinates $(x_1, y_1)$, $(x_2, y_1)$, $(x_1, y_2)$, and $(x_2, y_2)$. This schematic shows three sources for readability; the formulation supports an arbitrary number of sources.

**Table 1.** Responding functions of source/sink terms and their definitions.

| Source Function | Response Function | Definition of Response Function |
|---|---|---|
| $p_{x,i}^m(x)$ | $H\left(x - x_{p1}^m\right) - H\left(x - x_{p2}^m\right)$ | Indicates that contaminant source *m* is in the region $x_{p1}^m \leq x \leq x_{p2}^m$ in the x direction |
| $p_{y,i}^m(y)$ | $H\left(y - y_{p1}^m\right) - H\left(y - y_{p2}^m\right)$ | Indicates that contaminant source *m* is in the region $y_{p1}^m \leq y \leq y_{p2}^m$ in y direction |
| $q_i^m(t)$ | 1 | Indicates that contaminant source m is continuously injected |
|  | $H(t) - H\left(t - t_q^m\right)$ | Indicates that contaminant source *m* is injected during the time interval $0 \leq t \leq t_q^m$ |

It is assumed that no contaminant mass is present initially within the domain, as expressed by the following initial condition:

$$C_i(x, y, t = 0) = 0 \qquad i = 1, 2, \dots, \text{N} \tag{3}$$

$$S_i(x, y, t = 0) = 0 \qquad i = 1, 2, \ldots, N \tag{4}$$

The following boundary conditions are imposed to ensure a unique solution to Equation (1):

$$-D_L \frac{\partial C_i(x = 0, y, t)}{\partial x} + vC_i(x = 0, y, t) = vf_i(t)[H(y - y_1) - H(y - y_2)] \qquad i = 1, 2, \ldots, N \tag{5}$$

$$\frac{\partial C_i(x = L, y, t)}{\partial x} = 0 \qquad i = 1, 2, \ldots, N \tag{6}$$

$$\frac{\partial C_i(x, y = 0, t)}{\partial y} = 0 \qquad i = 1, 2, \ldots, N \tag{7}$$

$$\frac{\partial C_i(x, y = W, t)}{\partial y} = 0 \qquad i = 1, 2, \ldots, N \tag{8}$$

where $L$ and $W$ are the length and width of the transport system, respectively.

The nondimensional form of Equations (1) through (8) is given as follows:

$$\frac{1}{Pe_L} \frac{\partial^2 C_i(x_D, y_D, t_D)}{\partial x_D{}^2} - \frac{\partial C_i(x_D, y_D, t_D)}{\partial x_D} + \frac{\gamma^2}{Pe_T} \frac{\partial^2 C_i(x_D, y_D, t_D)}{\partial y_D{}^2} - \lambda_i C_i(x_D, y_D, t_D) + \lambda_{i-1} C_{i-1}(x_D, y_D, t_D) +$$
$$\sum_{m=1}^{NS} \Omega_i^m p_{x,i}^m(x_D) p_{y,i}^m(y_D) q_i^m(t_D) - \frac{B_i}{\phi}\left(C_i(x_D, y_D, t_D) - \frac{S_i(x_D, y_D, t_D)}{K_{di}}\right) = \frac{\partial C_i(x_D, y_D, t_D)}{\partial t_D} \qquad i = 1, 2, \ldots, N \tag{9}$$

$$\rho_b \frac{\partial S_i(x_D, y_D, t_D)}{\partial t_D} = B_i \left(C_i(x_D, y_D, t_D) - \frac{S_i(x_D, y_D, t_D)}{K_{di}}\right) \qquad i = 1, 2, \ldots, N \tag{10}$$

$$C_i(x_D, y_D, t_D = 0) = 0 \qquad i = 1, 2, \ldots, N \tag{11}$$

$$S_i(x_D, y_D, t_D = 0) = 0 \qquad i = 1, 2, \ldots, N \tag{12}$$

$$-\frac{1}{Pe_L} \frac{\partial C_i(x_D = 0, y_D, t_D)}{\partial x_D} + C_i(x_D = 0, y_D, t_D) = C_{i,0}[H(y_D - y_{D1}) - H(y_D - y_{D2})] \qquad i = 1, 2, \ldots, N \tag{13}$$

$$\frac{\partial C_i(x_D = 1, y_D, t_D)}{\partial x_D} = 0 \qquad i = 1, 2, \ldots, N \tag{14}$$

$$\frac{\partial C_i(x_D, y_D = 0, t_D)}{\partial y_D} = 0 \qquad i = 1, 2, \ldots, N \tag{15}$$

$$\frac{\partial C_i(x_D, y_D = 1, t_D)}{\partial y_D} = 0 \qquad i = 1, 2, \ldots, N \tag{16}$$

where $x_D = \frac{x}{L}$, $y_D = \frac{y}{W}$, $t_D = \frac{vt}{L}$, $Pe_L = \frac{vL}{D_L}$, $Pe_T = \frac{vL}{D_T}$, $\Omega_i^m = \frac{M_i^m}{\phi} \frac{L}{v}$, $B_i = \frac{\beta_i L}{v}$, $\lambda_i = \frac{k_i L}{v}$, $\gamma = \frac{L}{W}$.

Based on the governing equations, initial conditions, and boundary conditions defined above, this study employs a series of mathematical transformations to derive a semi-analytical solution for the two-dimensional multi-species advection–dispersion equation with internal sources. Specifically, the Laplace transform is applied to the temporal variable t, the finite Fourier cosine transform is used for the transverse spatial variable y, and the generalized integral transform is applied to the longitudinal spatial variable x. Through these transformations and subsequent inverse operations, the final semi-analytical solution— presented as Equation (A23)—is obtained. A detailed derivation process is provided in Appendix A.

## 3. Results and Discussion

### 3.1. Verification of the Novel Semi-Analytical Model

In developing the proposed semi-analytical model for multispecies contaminant transport, it is essential not only to derive the solution to the governing equations but also to

verify the model's accuracy to ensure its applicability to a wide range of real-world scenarios. For this purpose, a representative case study involving chlorinated solvents is employed as a verification example. Chlorinated solvents are widely recognized as typical multispecies contaminants, with perchloroethylene (PCE) serving as the parent compound that sequentially degrades under anaerobic conditions into trichloroethylene (TCE), dichloroethylene (DCE), vinyl chloride (VC), and eventually ethene (ETH). To validate the accuracy of the developed model, simulation results are compared against those obtained from the Laplace Transform Finite Difference (LTFD) method—a well-established numerical approach for solving transient reactive transport problems in porous media, as described by Moridis and Reddell [26] and Chen et al. [27]. The simulation scenario is adapted from the illustrative example provided in the BIOCHLOR user manual [1], ensuring that the model is tested against a benchmark problem representative of multispecies degradation behavior.

The model is verified using two scenarios involving multiple contaminant sources. In the first scenario, the sources release contaminants continuously at a constant rate. In the second scenario, contaminant injection occurs over discrete time intervals, represented by a unit step function: solutes are steadily introduced during a one-year period from $t = 0$ to $t = 1$ year. This dual-scenario design allows the model to capture both continuous and time-dependent contaminant input conditions. In both cases, the initial condition assumes a zero concentration throughout the domain, and no inlet boundary sources are considered. Each internal source releases perchloroethylene (PCE) at a concentration of 10 mg/L, serving as the parent compound in a sequential biodegradation chain. The degradation process follows first-order kinetics, producing trichloroethylene (TCE), dichloroethylene (DCE), vinyl chloride (VC), and ethene (ETH) as intermediate and final products. Model outputs are evaluated at $t = 2$ years, with the sorption rate uniformly set to 0.5 year$^{-1}$ for verification. The spatial distribution of contaminant sources is illustrated in Figure 2, and the complete set of model parameters is provided in Table 2.

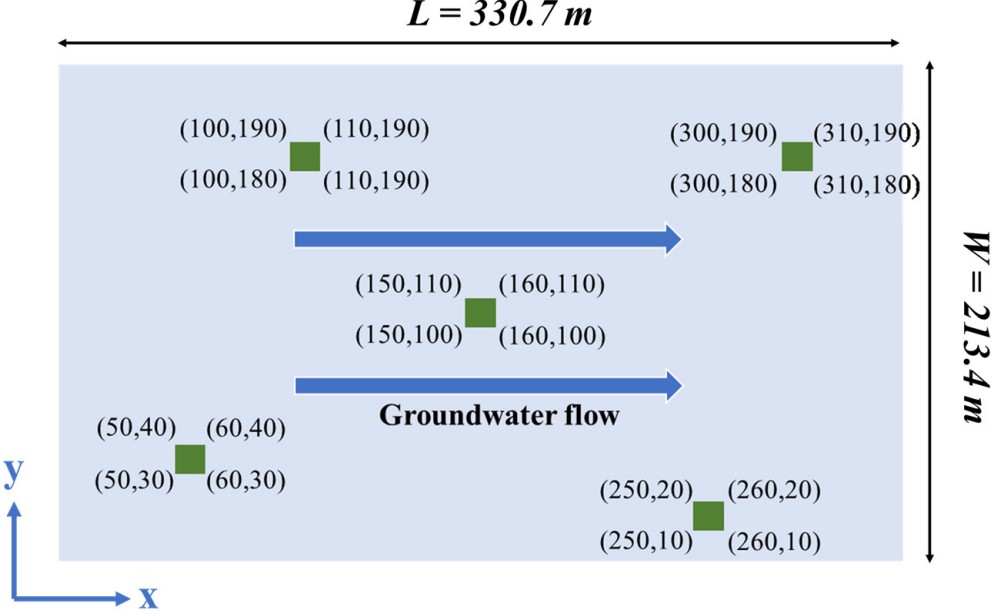

**Figure 2.** The conceptual model for verification includes five pollution sources located at arbitrary positions.

**Table 2.** Transport parameters used for model verification transport tetrachloroethene (PCE) and daughter products formed through natural degradation, including trichloroethylene (TCE), dichloroethane (DCE) isomers (1,1-DCE, cis 1, 2-DCE, and trans 1,2-DCE), vinyl chloride (VC), and ethene (ETH) in groundwater.

| Parameters | Value | | | | |
|---|---|---|---|---|---|
| Length, $L$ [m] | 330.7 | | | | |
| Width, $W$ [m] | 213.4 | | | | |
| Seepage velocity, $v$ [m year$^{-1}$] | 10 | | | | |
| Longitudinal dispersion coefficient, $D_L$ [m$^2$ year$^{-1}$] | 2000 | | | | |
| Transverse dispersion coefficient, $D_T$ [m$^2$ year$^{-1}$] | 200 | | | | |
| Effective porosity, $\phi$ [-] | 0.2 | | | | |
| Bulk density, $\rho_b$ [kg/L] | 1.6 | | | | |
| Simulation time, $t$ [year] | 2 | | | | |
| | PCE | TCE | DCE | VC | ETH |
| Distribution coefficient, $k_{di}$ [kg m$^{-3}$] | 0.784 | 0.239 | 0.230 | 0.0545 | 0.556 |
| First-order degradation reaction rate, $k_i$ [year$^{-1}$] | 2 | 1 | 0.7 | 0.4 | 0 |
| Sorption rate constant, $\beta_i$ [year$^{-1}$] | 0.5 | 0.5 | 0.5 | 0.5 | 0.5 |
| Contaminant Injected mass of sources, $M$ [mg/L] | 10 | 0 | 0 | 0 | 0 |

Figure 3 illustrates the spatial concentration distributions of all contaminants in the chlorinated solvent biodegradation chain for Scenario 1, while Figure 4 presents the corresponding results for Scenario 2. In both scenarios, the proposed semi-analytical model shows excellent agreement with the numerical model, demonstrating its accuracy in capturing multispecies transport behavior. These verification results confirm the model's capability to simulate the migration of both parent compounds and their degradation products from multiple contaminant sources simultaneously. Moreover, the consistency observed across scenarios underscores the robustness and practical applicability of the computer code developed to implement the semi-analytical solution.

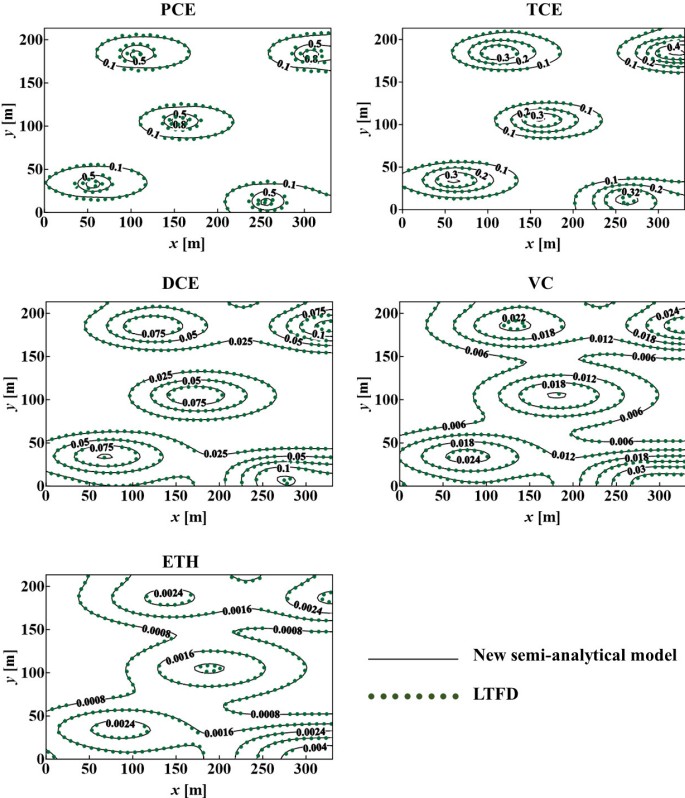

**Figure 3.** Comparison of the transport process of chlorinated solvent pollutants and their degradation products from multiple contaminant sources with constant injection between the new semi-analytical model and the corresponding numerical model.

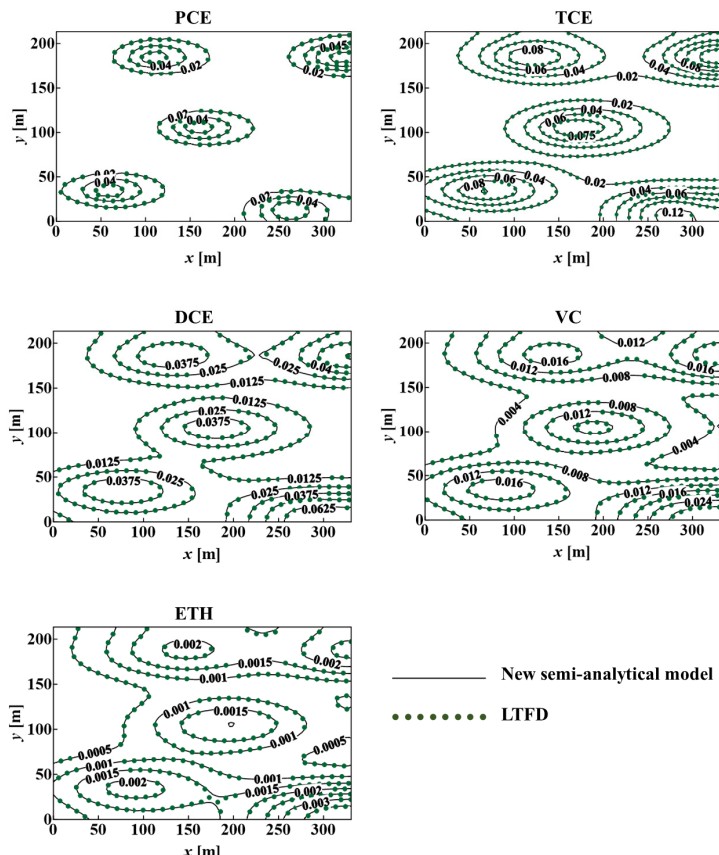

**Figure 4.** Comparison of the transport process of chlorinated solvents and their degradation products from multiple contaminant sources with pulse injection between the new semi-analytical model and the corresponding numerical model.

### 3.2. Effect of Sorption Rate on Dissolve Phase Concentration

This study investigates two scenarios involving different transport parameters for multiple contaminant sources. The inlet boundary source is assumed to remain constant over time and is spatially distributed between $y_1$ and $y_2$ (Figure 5). It contains all five species in the chlorinated solvent biodegradation chain, with an initial concentration of 1 mg/L assigned to each compound. Contaminant concentration profiles are analyzed along the longitudinal direction at an observation cross-section located at $y$ = 106.7 m. This transect intersects both the upstream inlet boundary source and the internal injection sources positioned at coordinates (150, 100), (150, 110), (160, 100), and (160, 110), respectively. Figures 6 and 7 illustrate the concentration distributions of the parent compound and its degradation products under varying sorption rates, considering both continuous and short-term injection scenarios. All simulation results correspond to $t$ = 2 years after the initiation of contaminant release into the groundwater system.

In the first scenario, concentration profiles for PCE, TCE, DCE, VC, and ETH are presented. The simulation results indicate that two years of continuous injection source, coupled with increasing sorption rates, leads to a gradual reduction in dissolved-phase contaminant concentrations. Sorption plays a significant role in limiting contaminant mobility and availability in the aqueous phase. As the sorption rate increases—from 0.05 year$^{-1}$ to 50 year$^{-1}$—a larger fraction of each contaminant becomes sorbed onto the solid matrix, resulting in lower concentrations in the dissolved phase. This behavior is clearly illustrated in Figure 6, where higher sorption rates ($\beta$) correspond to reduced peak concentrations and more rapid attenuation of contaminant levels with distance.

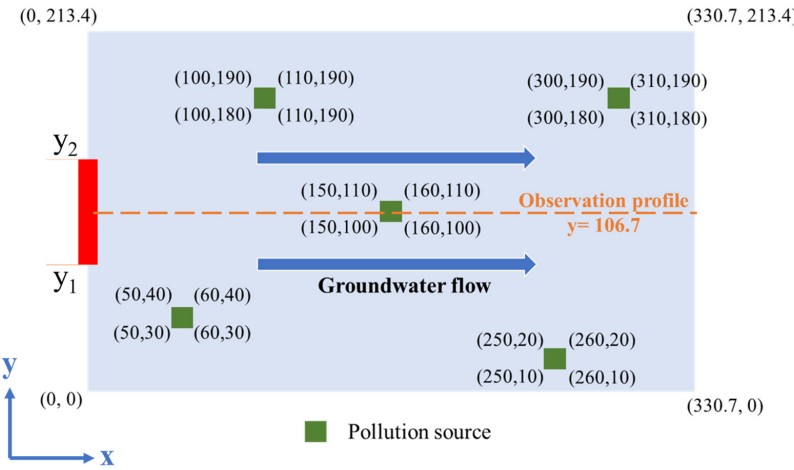

**Figure 5.** The conceptual model includes one main upstream source and five injected contaminant sources located at different locations. The red line indicates a one-dimensional concentration distribution observation profile.

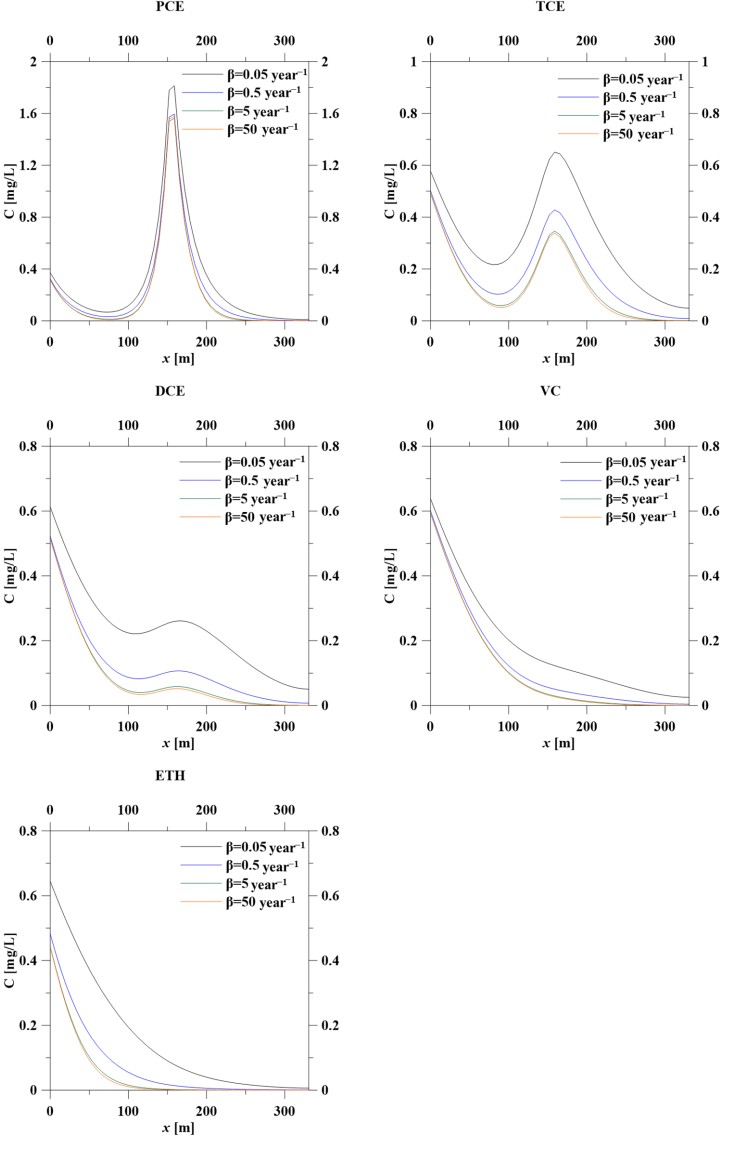

**Figure 6.** Observation profile at y = 106.7 m for the pollutants PCE, TCE, DCE, VC, and ETH, with both the main pollution source and injection sources continuous, under different sorption rates (β) over a simulation period of 2 years.

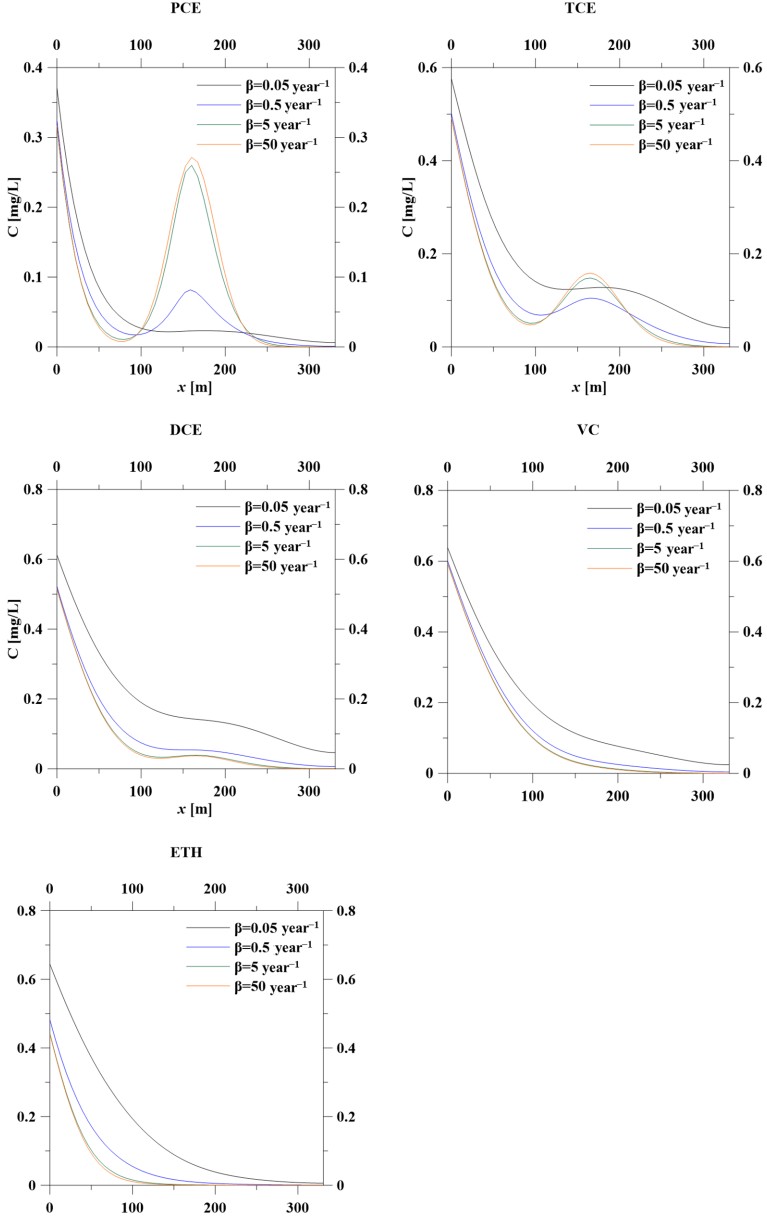

**Figure 7.** Observation profile at y = 106.7 m for the pollutants PCE, TCE, DCE, VC, and ETH, considering the primary contaminated source being continuous and other sources being pulse injected with an injection period of 1 year into the groundwater, under different sorption rates (β) over a simulation period of 2 years.

In the second scenario illustrated in Figure 7, short-term contaminant sources are represented using a Heaviside step function, simulating contaminant injection over a one-year period, while the inlet boundary sources are treated as continuous. As the sorption rate increases, the concentrations of both parent and daughter species generally decline downstream from the continuous upstream source, consistent with the expected effect of sorption in retarding contaminant migration. However, a contrasting trend is observed near the short-term injection source, particularly for PCE and TCE. At higher sorption rates (β = 5 and 50 year$^{-1}$), significantly elevated concentrations are predicted in the vicinity of the injection point. This phenomenon suggests that high sorption capacity may result in local accumulation due to rapid desorption, leading to pronounced peak concentrations near the source. For DCE, the impact of sorption is less pronounced but still evident, with slightly higher concentrations near the injection site observed at increased sorption rates. These results highlight the complex interplay between sorption kinetics and

contaminant transport behavior, especially under pulse-loading conditions. The emergence of local concentration peaks for PCE and TCE underscores the importance of accurately characterizing sorption–desorption dynamics when assessing and managing groundwater contamination from transient pollutant sources.

To better understand this behavior, the concentration of the sorbed phase at a fixed location is analyzed for the first species in the degradation chain, as shown in Figure 8. This figure compares the temporal variations in the concentrations of five species under two conditions: a continuous pollution source and a short-term injection source, both simulated at the same sorption rate of 50 year$^{-1}$. Under continuous source conditions, the concentrations of all species in the dissolved phase steadily increase over time, reflecting the sustained input of contaminants. In contrast, for the short-term injection scenario, PCE and TCE exhibit a rapid increase in concentration following the start of injection, followed by a sharp decline after the source is removed at $t = 1$ year. This pattern suggests that desorption plays a key role in sustaining elevated concentrations temporarily, even after the cessation of contaminant input.

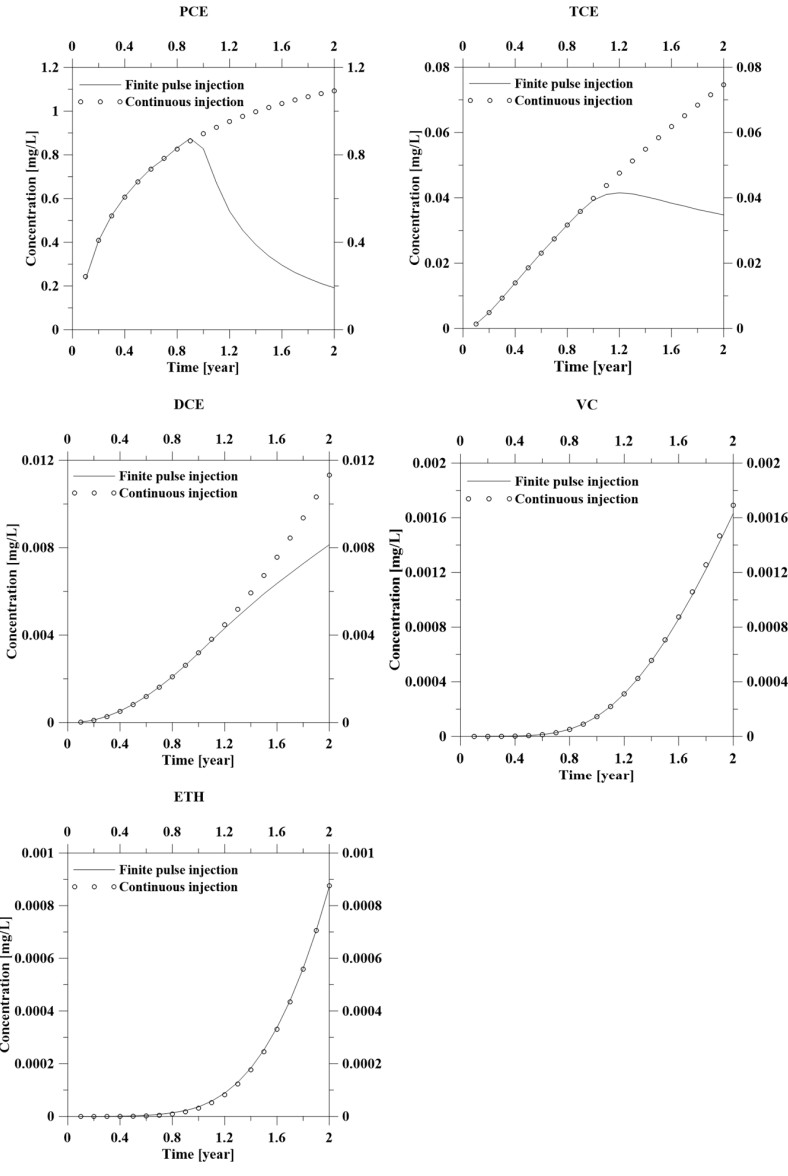

**Figure 8.** Comparison of concentration changes over time of pollutants belonging to the biodegradation chain of chlorinated solvents in the sorbed phase between pulse injection and continuous injection source with sorption rate, β = 50 years$^{-1}$.

In the continuous injection scenario, a sustained supply of PCE leads to the steady formation of degradation products. Over time, the exchange between the sorbed and dissolved phases approaches equilibrium, resulting in stabilized concentrations of these compounds in both phases. In contrast, the finite pulse injection scenario shows that when dissolved-phase concentrations decline due to the termination of the contaminant source, previously sorbed contaminants desorb back into the dissolved phase. This includes intermediate products such as DCE and VC, as well as the final product, ETH. However, due to the low concentrations of these compounds and the short simulation period, the desorption effect is less pronounced. Understanding these dynamics is essential for accurately predicting contaminant behavior and designing effective remediation strategies that address both the parent compound and its degradation products to ensure comprehensive environmental protection.

## 4. Conclusions

This study develops a semi-analytical model for simulating multispecies contaminant transport in groundwater systems with multiple internal sources under rate-limited sorption conditions. The model accounts for key transport processes, including advection, dispersion, first-order degradation, and rate-limited sorption. To solve the governing equations, a hybrid analytical approach combining the Laplace transform, finite cosine Fourier transform, and generalized integral transform is employed, followed by inverse transformations to obtain time-domain solutions. This solution framework enables efficient and accurate simulation of two-dimensional multispecies transport driven by complex, time-dependent internal sources. Model validation was carried out through scenarios involving five internal pollution sources under both continuous and pulse injection modes. The contaminant chain consists of chlorinated solvents, beginning with the parent compound PCE and sequentially degrading to TCE, DCE, VC, and finally ETH. The high consistency between the semi-analytical and numerical results confirms the model's accuracy and computational efficiency. The study further explores the effects of rate-limited sorption on the migration and distribution of both parent and daughter species. For continuously injected sources, higher sorption rates reduce dissolved-phase concentrations due to increased retention in the solid phase. Conversely, for pulse-injected sources, desorption becomes dominant after source cessation, leading to elevated dissolved-phase concentrations—especially for PCE.

This 2-D semi-analytical model is computationally efficient and easy to apply. It successfully handles multiple contaminant sources under rate-limited sorption conditions and demonstrates strong agreement with a standard numerical solver. However, it is designed for systems with uniform, steady flow and simple boundaries. Cases with strong heterogeneity or complex boundary conditions fall outside its intended scope. By integrating a rigorous semi-analytical solution method with realistic sorption kinetics and internal source configurations, the proposed model offers a robust and flexible framework for predicting multispecies contaminant transport. It facilitates the evaluation of diverse pollution scenarios and supports the development of more effective, site-specific remediation strategies.

**Author Contributions:** Conceptualization, J.-S.C.; methodology, validation, writing—original draft preparation, T.-U.N. and Y.-H.C.; writing—review and editing, H.S., C.-P.L., and J.-S.C.; supervision, J.-S.C. All authors have read and agreed to the published version of the manuscript.

**Funding:** This research received no external funding.

**Data Availability Statement:** No new data were created or analyzed in this study. Data sharing does not apply to this article.

**Conflicts of Interest:** The authors declare no conflicts of interest.

## Appendix A

In the development of analytical solution models, the Laplace transform has been commonly employed to eliminate the time-dependent differential term. The Laplace transform of the concentration $C_i(x_D, y_D, t_D)$ is defined as

$$\overline{C_i}(x_D, y_D, s) = L[C_i(x_D, y_D, t_D)] = \int_0^\infty e^{-st_D} C_i(x_D, y_D, t_D) dt_D$$

where s is the Laplace conversion parameter of t. The Laplace inverse transformation formula can be expressed as

$$C_i(x_D, y_D, t_D) = L^{-1}\left[\overline{C_i}(x_D, y_D, s)\right] = \frac{1}{2\pi i}\int_{\alpha-i\infty}^{\alpha+i\infty} e^{st_D}\overline{C_i}(x_D, y_D, s) ds$$

where $\alpha$ is the abscissa value of the convergence interval of the function.

By applying the Laplace transform to Equations (9)–(16) using the definitions above, the equations can be reformulated as follows:

$$
\frac{1}{Pe_L}\frac{\partial^2 \overline{C_i}(x_D, y_D, s)}{\partial x_D^2} - \frac{\partial \overline{C_i}(x_D, y_D, s)}{\partial x_D} + \frac{\gamma^2}{Pe_T}\frac{\partial^2 \overline{C_i}(x_D, y_D, s)}{\partial y_D^2}
$$
$$
-(s+\lambda_i)\overline{C_i}(x_D, y_D, s) + \lambda_{i-1}\overline{C_{i-1}}(x_D, y_D, s) - \frac{B_i}{\phi}\left(\overline{C_i}(x_D, y_D, s) - \frac{\overline{S_i}(x_D, y_D, s)}{K_{di}}\right)
$$
$$
= -\sum_{m=1}^{NS}\Omega_i^m p_{x,i}^m(x_D)p_{y,i}^m(y_D)Q_i^m(s) \qquad i = 1, 2, \ldots, N
\tag{A1}
$$

$$
s\rho_b\overline{S_i}(x_D, y_D, s) = B_i\left(\overline{C_i}(x_D, y_D, s) - \frac{\overline{S_i}(x_D, y_D, s)}{K_{di}}\right) \qquad i = 1, 2, \ldots, N
\tag{A2}
$$

$$
\overline{C_i}(x_D, y_D, s=0) = 0 \qquad \overline{S_i}(x_D, y_D, s=0) = 0 \qquad i = 1, 2, \ldots, N
\tag{A3}
$$

$$
-\frac{1}{Pe_L}\frac{\partial \overline{C_i}(x_D=0, y_D, s)}{\partial x_D} + \overline{C_i}(x_D=0, y_D, s) = \frac{C_{i,0}}{s}[H(y_D - y_{D1}) - H(y_D - y_{D2})] \qquad i = 1, 2, \ldots, N
\tag{A4}
$$

$$
\frac{\partial \overline{C_i}(x_D=1, y_D, s)}{\partial x_D} = 0 \qquad i = 1, 2, \ldots, N
\tag{A5}
$$

$$
\frac{\partial \overline{C_i}(x_D, y_D=0, s)}{\partial y_D} = 0 \qquad i = 1, 2, \ldots, N
\tag{A6}
$$

$$
\frac{\partial \overline{C_i}(x_D, y_D=1, s)}{\partial y_D} = 0 \qquad i = 1, 2, \ldots, N
\tag{A7}
$$

Here, $Q_i^m(s)$ is the function after Laplace transformation of $q_i^m(t_D)$, expressed as follows:

$$
Q_i^m(s) = \int_0^\infty e^{-st_D} q_i^m(t_D) dt_D = \begin{cases} \frac{1}{s} & , if\ q_i^m(t_D) = 1 \\ \frac{1}{s}\left(1 - e^{-st_q^m}\right) & , if\ q_i^m(t_D) = H(t) - H(t - t_q^m) \end{cases}
$$

Substituting the Laplace-converted rate-limited sorption relationship in Equation (A2) into the governing Equation (A1), it can be expressed as

$$
\frac{1}{Pe_L}\frac{\partial^2 \overline{C_i}(x_D, y_D, s)}{\partial x_D^2} - \frac{\partial \overline{C_i}(x_D, y_D, s)}{\partial x_D} + \frac{\gamma^2}{Pe_T}\frac{\partial^2 \overline{C_i}(x_D, y_D, s)}{\partial y_D^2} - \Theta_i(s)\overline{C_i}(x_D, y_D, s) + \lambda_{i-1}\overline{C_{i-1}}(x_D, y_D, s) =
$$
$$
-\sum_{m=1}^{NS}\Omega_i^m p_{x,i}^m(x_D)p_{y,i}^m(y_D)Q_i^m(s) \qquad i = 1, 2, \ldots, N
\tag{A8}
$$

where $\Theta_i(s) = s + \lambda_i + \frac{B_i}{\phi}\frac{sK_{di}\rho_b}{sK_{di}\rho_b + B_i}$

The following is the definition of finite Fourier cosine conversion:

$$
H_i(x_D, n, s) = \int_0^1 \overline{C_i}(x_D, y_D, s)\cos(n\pi y_D) dy_D \qquad i = 1, 2, \ldots, N
\tag{A9}
$$

Here $H_i(x_D, n, s)$ is the dissolved-phase concentration after finite Fourier cosine conversion; n is the finite Fourier cosine conversion parameter of y.

The inverse conversion formula of finite Fourier cosine can be expressed as

$$\overline{C}_i(x_D, y_D, s) = H_i(x_D, n = 0, s) + 2\sum_{n=1}^{\infty} H_i(x_D, n, s)cos(n\pi y_D) \qquad i = 1, 2, \ldots, N \quad (A10)$$

Through the finite Fourier cosine transformation of Equations (A4)–(A8) through the above definitions, the governing equations and boundary conditions can be rewritten as

$$\frac{1}{Pe_L}\frac{d^2 H_i(x_D, n, s)}{dx_D^2} - \frac{dH_i(x_D, n, s)}{dx_D} - \left(\Theta_i(s) + \frac{\gamma^2 n^2 \pi^2}{Pe_T}\right)H_i(x_D, n, s) + \lambda_{i-1}H_{i-1}(x_D, n,) =$$
$$-\sum_{m=1}^{NS} \Omega_i^m p_{x,i}^m(x_D)P_{y,i}^m(n)Q_i^m(s) \qquad i = 1, 2, \ldots, N \quad (A11)$$

$$-\frac{1}{Pe_L}\frac{\partial H_i(x_D = 0, n, s)}{\partial x_D} + H_i(x_D = 0, n, s) = \frac{C_{i,0}}{s}\Phi(n) \qquad i = 1, 2, \ldots, N \quad (A12)$$

$$\frac{\partial H_i(x_D = 1, n, s)}{\partial x_D} = 0 \qquad i = 1, 2, \ldots, N \quad (A13)$$

$\Phi(n)$ in boundary condition Equation (A12) is a function of $H(y_D - y_{D1}) - H(y_D - y_{D2})$ after finite Fourier cosine transformation, and its value is expressed as follows:

$$\Phi(n) = \begin{cases} y_{D2} - y_{D1} \; n = 0 \\ \frac{sin(n\pi y_{D2}) - sin(n\pi y_{D1})}{n\pi} \; n = 1, 2, 3\ldots \end{cases}$$

In the past, Pérez Guerrero et al. [18] established a method: using variable transformation to homogenize the boundary conditions and eliminate the first-order space differential term in the x direction in the control equation, to facilitate subsequent generalized Integral transformations that can directly convert governing equations into algebraic equations. This study defines the variable transformation function as follows, based on the derivation steps of [20]:

$$H_i(x_D, n, s) = \frac{C_{i,0}}{s}\Phi(n) + e^{\frac{Pe_L}{2}x_D}\varphi_i(x_D, n, s)$$

Here, $\varphi_i(x_D, n, s)$ is the dissolved phase concentration after variable transformation. By transforming Equations (A11)–(A13), the governing equations and boundary conditions can be rewritten as

$$\frac{1}{Pe_L}\frac{d^2 \varphi_i(x_D, n, s)}{dx_D^2} - \left(\Theta_i(s) + \frac{\gamma^2 n^2 \pi^2}{Pe_T} + \frac{Pe_L}{4}\right)\varphi_i(x_D, n, s)$$
$$= \left(\Theta_i(s) + \frac{\gamma^2 n^2 \pi^2}{Pe_T}\right)\frac{C_{i,0}}{s}\Phi(n)e^{-\frac{Pe_L}{2}x_D} - \lambda_{i-1}\left(\frac{C_{i-1,0}}{s}\Phi(n)e^{-\frac{Pe_L}{2}x_D} + \varphi_{i-1}(x_D, n, s)\right) \quad (A14)$$
$$-\sum_{m=1}^{NS} \Omega_i^m p_{x,i}^m(x_D)P_{y,i}^m(n)Q_i^m(s)e^{-\frac{Pe_L}{2}x_D} \qquad i = 1, 2, \ldots, N$$

$$-\frac{d\varphi_i(x_D = 0, n, s)}{dx_D} + \frac{Pe_L}{2}\varphi_i(x_D = 0, n, s) = 0 \qquad i = 1, 2, \ldots, N \quad (A15)$$

$$\frac{d\varphi_i(x_D = 1, n, s)}{dx_D} + \frac{Pe_L}{2}\varphi_i(x_D = 1, n, s) = 0 \qquad i = 1, 2, \ldots, N \quad (A16)$$

This study employs a generalized integral conversion method based on the approach of [15], focusing on a structured four-step process. First, the eigenvalue problem is addressed by determining the eigenvalues, eigenfunctions, norms, and normalized eigenfunctions (kernel function). Next, a generalized integral transformation and its inverse are defined, facilitating the conversion of ordinary differential equations (ODEs) into algebraic equations. The transformed ODEs are then solved in this simplified algebraic form. Finally, the solution is reverted to the original domain using the inverse transformation and accumulation process, allowing the recovery of the unknown function in the original

problem. This method streamlines the process of solving complex differential equations by transforming them into a more manageable form.

The eigenvalue problem can be defined as follows, according to the governing Equation (A14) and boundary conditions (A15) and (A16):

$$\frac{d^2 K(x_D)}{dx_D^2} + \xi_l^2 K(x_D) = 0$$

$$\frac{dK(x_D = 0)}{dx_D} - \frac{Pe_L}{2} K(x_D = 0) = 0$$

$$\frac{dK(x_D = 1)}{dx_D} + \frac{Pe_L}{2} K(x_D = 1) = 0$$

Here, $K(x_D)$ is the characteristic function and $\xi_l$ is the generalized integral transformation parameter.

Solving the selected eigenvalue problem can obtain the normalized eigenfunction required for generalized integral transformation, which is expressed as follows:

$$K(\xi_l, x_D) = \frac{Pe_L}{2} sin(\xi_l x_D) + \xi_l cos(\xi_l x_D)$$

$\xi_l$ can be defined by the following formula: $\xi_l cot\xi_l - \frac{\xi_l^2}{Pe_L} + \frac{Pe}{4} = 0$

The generalized integral defined by the above normalized characteristic function is transformed into:

$$\overline{\varphi_i}(\xi_l, n, s) = \int_0^1 K(\xi_l, x_D)\varphi_i(x_D, n, s)dx_D \qquad i = 1, 2, \ldots, N \tag{A17}$$

where $\overline{\varphi_i}(\xi_l, n, s)$ is the dissolved phase concentration after generalized integral conversion. The generalized integral inverse transformation is defined as

Where $N(\xi_l)$ represents the reciprocal of the norm, which is defined as follows:

$$N(\xi_l) = \frac{2}{\frac{Pe_L^2}{4} + Pe_L + \xi_l^2}$$

By applying a generalized integral transformation to the governing Equation (A14), it can be reduced to an algebraic equation for further solution. The transformed equation is expressed as follows:

$$-\left(\Theta_i(s) + \frac{\gamma^2 n^2 \pi^2}{Pe_T} + \frac{Pe_L}{4} + \frac{\xi_l^2}{Pe_L}\right)\overline{\varphi_i}(\xi_l, n, s) = \left(\Theta_i(s) + \frac{\gamma^2 n^2 \pi^2}{Pe_T}\right)\frac{C_{i,0}}{s}\Phi(n)\Psi(\xi_l) - \lambda_{i-1}\left(\frac{C_{i-1,0}}{s}\Phi(n)\Psi(\xi_l) + \right.$$
$$\overline{\varphi_{i-1}}(\xi_l, n, s)) - \sum_{m=1}^{NS} \Omega_i^m P_{x,i}^m(\xi_l) P_{y,i}^m(n) Q_i^m(s) \qquad i = 1, 2, \ldots, N \tag{A18}$$

where $P_{x,i}^m(\xi_l)$ and $\Psi(\xi_l)$ is the function after generalized integral transformation of $p_{x,i}^m(x_D)$ and $e^{-\frac{Pe_L}{2}x_D}$, which are defined as follows:

$$P_{x,i}^m(\xi_l) = \int_0^1 K(\xi_l, x_D)p_{x,i}^m(x_D)e^{-\frac{Pe_L}{2}x_D}dx_D \qquad i = 1, 2, \ldots, N \tag{A19}$$

$$\Psi(\xi_l) = \int_0^1 K(\xi_l, x_D)e^{-\frac{Pe_L}{2}x_D}dx_D = \frac{Pe_L\xi_l}{\frac{Pe_L^2}{4} + \xi_l^2} \qquad i = 1, 2, \ldots, N \tag{A20}$$

To organize complex equations, let $\alpha_i(\xi_l, s)$ and $\delta(\xi_l)$ be

$$\alpha_i(\xi_l, s) = \Theta_i(s) + \frac{\gamma^2 n^2 \pi^2}{Pe_T} + \frac{Pe_L}{4} + \frac{\xi_l^2}{Pe_L}$$

$$\delta(\xi_l) = \frac{Pe_L}{4} + \frac{\xi_l^2}{Pe_L}$$

Arrange Equation (A18) and rewrite it as follows:

$$\overline{\varphi}_i(\xi_l, n, s) = -\frac{\alpha_i(\xi_l, s) - \delta(\xi_l)}{\alpha_i(\xi_l, s)} \frac{C_{i,0}}{s} \Phi(n) \Psi(\xi_l) + \frac{\lambda_{i-1}}{\alpha_i(\xi_l, s)} \left( \frac{C_{i-1,0}}{s} \Phi(n) \Psi(\xi_l) + \overline{\varphi}_{i-1}(\xi_l, n, s) \right) +$$
$$\sum_{m=1}^{NS} \Omega_i^m P_{x,i}^m(\xi_l) P_{y,i}^m(n) \frac{Q_i^m(s)}{\alpha_i(\xi_l, s)} \qquad i = 1, 2, \ldots, N \tag{A21}$$

Solving the algebraic Equation (A21) yields the dissolved phase concentrations of individual species, from which a general solution applicable to any number of species can be formulated:

$$\overline{\varphi}_i(\xi_l, n, s) = \left[ -\frac{C_{i,0}}{s} \frac{\alpha_i(\xi_l, s) - \delta(s)}{\alpha_i(\xi_l, s)} + \sum_{j=0}^{j=i-2} \frac{C_{i-j-1,0}}{s} \frac{\left( \prod_{k_1=0}^{k_1=j} \lambda_{i-k_1-1} \right) \delta(s)}{\prod_{k_2=0}^{k_2=j+1} \alpha_{i-k_2}(\xi_l, s)} \right] \Phi(n) \Psi(\xi_l) +$$
$$\sum_{m=1}^{NS} \left[ \begin{array}{l} \Omega_i^m P_{x,i}^m(\xi_l) P_{y,i}^m(n) \frac{Q_i^m(s)}{\alpha_i(\xi_l, s)} + \\ \sum_{k=0}^{k=i-2} \frac{\prod_{j_1=0}^{j_1=k} \lambda_{i-j_1-1}}{\prod_{j_2=0}^{j_2=k} \alpha_{i-j_2}(\xi_l, s)} \Omega_{i-k-1}^m P_{x,i-k-1}^m(\xi_l) P_{y,i-k-1}^m(n) \frac{Q_{i-k-1}^m(s)}{\alpha_{i-k-1}(\xi_l, s)} \end{array} \right] \quad i = 1, 2, \ldots, N \tag{A22}$$

This study applies the generalized integral inverse transformation, variable transformation, and finite Fourier cosine inverse transformation to derive a semi-analytical solution of the algebraic Equation (A22) in the transform domain, resulting in the final form presented in Equation (A23) with $i = 1, 2, 3, \ldots, N$.

$$\overline{C}_i(x_D, y_D, s) = \frac{C_{i,0}}{s} \Phi(n = 0) + e^{\frac{Pe_L}{2} x_D} \sum_{l=1}^{\infty} K(\xi_l, x_D) N(\xi_l)$$
$$\times \left[ \begin{array}{l} \left[ -\frac{C_{i,0}}{s} \frac{\alpha_i(\xi_l, s) - \delta(s)}{\alpha_i(\xi_l, s)} + \sum_{j=0}^{j=i-2} \frac{C_{i-j-1,0}}{s} \frac{\left( \prod_{k_1=0}^{k_1=j} \lambda_{i-k_1-1} \right) \delta(s)}{\prod_{k_2=0}^{k_2=j+1} \alpha_{i-k_2}(\xi_l, s)} \right] \Phi(n) \Psi(\xi_l) \\ + \sum_{m=1}^{NS} \left[ \begin{array}{l} \Omega_i^m P_{x,i}^m(\xi_l) P_{y,i}^m(n) \frac{Q_i^m(s)}{\alpha_i(\xi_l, s)} \\ + \sum_{k=0}^{k=i-2} \frac{\prod_{j_1=0}^{j_1=k} \lambda_{i-j_1-1}}{\prod_{j_2=0}^{j_2=k} \alpha_{i-j_2}(\xi_l, s)} \Omega_{i-k-1}^m P_{x,i-k-1}^m(\xi_l) P_{y,i-k-1}^m(n) \frac{Q_{i-k-1}^m(s)}{\alpha_{i-k-1}(\xi_l, s)} \end{array} \right] \end{array} \right]$$
$$+ 2 \sum_{n=1}^{\infty} \left( \begin{array}{l} \frac{C_{i,0}}{s} \Phi(n) + e^{\frac{Pe_L}{2} x_D} \sum_{l=1}^{\infty} K(\xi_l, x_D) N(\xi_l) \\ \times \left\{ \begin{array}{l} \left[ -\frac{C_{i,0}}{s} \frac{\alpha_i(\xi_l, s) - \delta(s)}{\alpha_i(\xi_l, s)} + \sum_{j=0}^{j=i-2} \frac{C_{i-j-1,0}}{s} \frac{\left( \prod_{k_1=0}^{k_1=j} \lambda_{i-k_1-1} \right) \delta(s)}{\prod_{k_2=0}^{k_2=j+1} \alpha_{i-k_2}(\xi_l, s)} \right] \Phi(n) \Psi(\xi_l) \\ + \sum_{m=1}^{NS} \left[ \begin{array}{l} \Omega_i^m P_{x,i}^m(\xi_l) P_{y,i}^m(n) \frac{Q_i^m(s)}{\alpha_i(\xi_l, s)} \\ + \sum_{k=0}^{k=i-2} \frac{\prod_{j_1=0}^{j_1=k} \lambda_{i-j_1-1}}{\prod_{j_2=0}^{j_2=k} \alpha_{i-j_2}(\xi_l, s)} \Omega_{i-k-1}^m P_{x,i-k-1}^m(\xi_l) P_{y,i-k-1}^m(n) \frac{Q_{i-k-1}^m(s)}{\alpha_{i-k-1}(\xi_l, s)} \end{array} \right] \end{array} \right\} \end{array} \right)$$
$$\times \cos(n\pi y_D) \tag{A23}$$
$$i = 1 \ldots N$$

A computer program to simulate multi-species pollutant transport and predict contaminant concentrations is developed in the FORTRAN language, based on the general Equation (A23).

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
