# Peer review of "Development of a Semi-Analytical Solution for Simulating the Migration of Parent and Daughter Contaminants from Multiple Contaminant Sources, Considering Rate-Limited Sorption Effects"

_hydrology, doi:10.3390/hydrology12100249_

Round 1

Reviewer 1 Report

Comments and Suggestions for Authors From a technical standpoint, this is a very well-written paper manuscript. It is easy to follow.
However, from the perspective of research, the "Development of a Semi-Analytical Solution for Simulating the Migration of Parent and Daughter Contaminants" is a highly outdated method. The scientific value of similar research may be limited. Therefore, few researchers are likely to be interested in this type of research.
Below are three specific comments:
1) The abstract is written in a very simple and incomplete manner. I recommend adding the main research findings, such as improvements in simulation accuracy.
2) Figure 1. Please add a legend and scale. I am unclear about the numerical values represented in the figure. Are the units in meters?
3) Please add a subsection to the discussion, including the advantages and disadvantages of this method, how the results compare to numerical simulation methods, and under what conditions this two-dimensional computational model can be applied.

Author Response

Response to Reviewer 1 Comments

1. Summary

We would like to sincerely thank reviewer for your time, thoughtful comments, and constructive suggestions. We have carefully considered all feedback and made corresponding revisions to the manuscript. All changes have been marked using red font in the revised version, as per the journal’s submission guidelines.

Our point-by-point responses to each reviewer's comment are provided below. Where applicable, we have indicated the specific page, paragraph, and line number in the revised manuscript where changes have been made.

2. Questions for General Evaluation

Reviewer’s Evaluation

Response and Revisions

Does the introduction provide sufficient background and include all relevant references?

Are all the cited references relevant to the research?

Is the research design appropriate?

Are the methods adequately described?

Are the results clearly presented?

Are the conclusions supported by the results?

3. Point-by-point response to Comments and Suggestions for Authors

Comment 1: The abstract is written in a very simple and incomplete manner. I recommend adding the main research findings, such as improvements in simulation accuracy.

Response 1:

Thank you for this valuable suggestion.

We have revised the Abstract to more clearly highlight the paper’s key contributions, particularly the novelty of the multi-species, multi-source model, with all changes marked in the revised manuscript.

Comment 2: Figure 1. Please add a legend and scale. I am unclear about the numerical values represented in the figure. Are the units in meters?

Response 2:

Thank you for your helpful comment. We apologize for the lack of clarity.

In the original manuscript, Figure 1 depicted the verification setup; the numbers shown were the source corner coordinates expressed in meters. In the revision, Figure 1 now includes a complete legend and axis labels in meters, and the caption explicitly states these units. To avoid confusion with the verification case, we have also added a generalized conceptual schematic in the Mathematical Model section, where all symbols are explained. Details are presented in the revised manuscript.

Comment 3: Please add a subsection to the discussion, including the advantages and disadvantages of this method, how the results compare to numerical simulation methods, and under what conditions this two-dimensional computational model can be applied.

Response 3:  

Thank you for the helpful suggestion.

We have addressed this by revising the Conclusion section to clearly state the advantages, limitations, and applicability of the proposed semi-analytical model. Specifically, we now emphasize that the model handles multiple internal sources under rate-limited sorption and shows strong agreement with a standard numerical solver. We also clarify that the model is designed for systems with uniform, steady flow and simple boundaries, and that cases with strong heterogeneity or complex boundary conditions fall outside its intended scope. The revised conclusion provides a clear summary of when and how the model can be effectively applied.

These amendments have been incorporated in lines 343 to 345 of the revised manuscript.

Reviewer 2 Report

Comments and Suggestions for Authors

Development of a Semi-Analytical Solution for Simulating the Migration of Parent and Daughter Contaminants from Multiple Contaminant Sources, Considering Rate-Limited Sorption Effects

Abstract:

Line 17: involving multiple contaminant sources

Line 26: sensitive to the variations in time-dependent sources

Introduction:

In the introduction, authors stated that “This study introduces a novel semi-analytical model for simulating the transport of multi-species contaminants in groundwater systems, explicitly accounting for rate-limited sorption and multiple time-dependent internal sources distributed arbitrarily within the domain. Unlike conventional analytical models that typically assume boundary-driven inputs, the proposed model incorporates source/sink terms into the coupled ADEs to better reflect real-world scenarios involving spatially distributed contamination events. To derive the solution efficiently, the governing equations are transformed into a system of algebraic equations using the Laplace transform in the time domain and integral transforms in the spatial dimensions. This modeling framework enables more realistic simulation of contaminant plume migration and transformation, particularly for degradable pollutants and their daughter products in complex subsurface environments”. It is not clear the limitations of the conventional analytical models except “assume boundary-driven inputs”, it needs more justification of this study and what are the shortages with “assume boundary-driven inputs”. It sees that the goal of this study is method development.

However, before this paragraph, authors cite and talk about single source contamination and it is important to simulate multiple source contaminations . . . is the goal of study “developing of models simulating multiple source contaminations”?

Additionally, in the Paragraph 5 of the introduction, authors stated “analytical solutions incorporating multiple internal sources remain scarce due to the increased mathematical complexity compared to traditional boundary-driven models. The development of such models is thus urgently needed to improve the applicability and accuracy of analytical solutions in real-world groundwater contamination assessments”. However, in the study case, authors compared results from their semi-analytical model to conventional analytical models (two scenarios presented), it is confusing to understand what is needed in the literature and what is the goal of this study?

Model developments include: 1. verification; calibration and validation

Authors mixed verification with validation.

In the case (two scenarios) authors presented, it is not clear the amount of chemicals released to the environment.

The reviewer suggests to change section 2 as:

  1. Methods and Procedure

2.1 Mathematical model

2.2 Model verification

A lot of stuff in section 3.1. belongs to 2.2. 

The specific improvements should the authors consider regarding the methodology?

Following are possible improvements I suggest:

  1. Validation of the model with real-world data
  2. Sensitivity analysis as well as uncertainty analysis
  3. Scalability and Computational Efficiency
  4. Application of the method with other contaminants

    Please also see my comments marked on the manuscript directly.

Author Response

Response to Reviewer 2 Comments

1. Summary

We would like to sincerely thank reviewer for your time, thoughtful comments, and constructive suggestions. We have carefully considered all feedback and made corresponding revisions to the manuscript. All changes have been marked using red font in the revised version, as per the journal’s submission guidelines.

Our point-by-point responses to each reviewer's comment are provided below. Where applicable, we have indicated the specific page, paragraph, and line number in the revised manuscript where changes have been made.

2. Questions for General Evaluation

Reviewer’s Evaluation

Response and Revisions

Does the introduction provide sufficient background and include all relevant references?

Must be improved

Are all the cited references relevant to the research?

Is the research design appropriate?

Must be improved

Are the methods adequately described?

Must be improved

Are the results clearly presented?

Can be improved

Are the conclusions supported by the results?

Must be improved

3. Point-by-point response to Comments and Suggestions for Authors

Comments 1: Abstract:

Line 17: involving multiple contaminant sources

Line 26: sensitive to the variations in time-dependent sources.

Response 1:

Thank you for your suggestion. We have revised the abstract according to your comments.

Comments 2: Introduction:

In the introduction, authors stated that “This study introduces a novel semi-analytical model for simulating the transport of multi-species contaminants in groundwater systems, explicitly accounting for rate-limited sorption and multiple time-dependent internal sources distributed arbitrarily within the domain. Unlike conventional analytical models that typically assume boundary-driven inputs, the proposed model incorporates source/sink terms into the coupled ADEs to better reflect real-world scenarios involving spatially distributed contamination events. To derive the solution efficiently, the governing equations are transformed into a system of algebraic equations using the Laplace transform in the time domain and integral transforms in the spatial dimensions. This modeling framework enables more realistic simulation of contaminant plume migration and transformation, particularly for degradable pollutants and their daughter products in complex subsurface environments”. It is not clear the limitations of the conventional analytical models except “assume boundary-driven inputs”, it needs more justification of this study and what are the shortages with “assume boundary-driven inputs”. It sees that the goal of this study is method development.

However, before this paragraph, authors cite and talk about single source contamination and it is important to simulate multiple source contaminations . . . is the goal of study “developing of models simulating multiple source contaminations”?

Additionally, in the Paragraph 5 of the introduction, authors stated “analytical solutions incorporating multiple internal sources remain scarce due to the increased mathematical complexity compared to traditional boundary-driven models. The development of such models is thus urgently needed to improve the applicability and accuracy of analytical solutions in real-world groundwater contamination assessments”. However, in the study case, authors compared results from their semi-analytical model to conventional analytical models (two scenarios presented), it is confusing to understand what is needed in the literature and what is the goal of this study?

Response 2:

Thank you for your valuable comment. We agree that the limitations of boundary-driven analytical models should be more clearly stated. Conventional analytical models often assume contaminant input from domain boundaries, which simplifies the mathematics but limits their applicability. In many real-world cases, however, contaminants originate from leaking tanks, landfills, or buried waste located within the domain. These internal sources are spatially distributed and often release contaminants at different times and positions, which cannot be easily captured by boundary-driven models.

In addition, most existing analytical models that include internal sources are limited to a single source location. Contaminated sites often involve multiple independent source zones, each with its own geometry and timing. Our model addresses this gap by allowing multiple time-dependent internal sources located arbitrarily within the domain. This enhances the model’s realism and expands its applicability to complex field conditions.

In the two verification scenarios, we compare the proposed semi-analytical model with a numerical solution based on the Laplace Transform Finite Difference (LTFD) method. Although the LTFD solver is based on the same conceptual framework as the semi-analytical solution following Moridis and Reddell (1991) and Chen et al. (1999), it is still a numerical method, not a fully analytical one. The comparison is conducted to verify the accuracy of our model, not to suggest that the existing analytical models already capture multiple internal sources.

Comments 3: Model developments include: 1. verification; calibration and validation

Authors mixed verification with validation.

In the case (two scenarios) authors presented, it is not clear the amount of chemicals released to the environment.

The reviewer suggests to change section 2 as:

Methods and Procedure

2.1 Mathematical model

2.2 Model verification

A lot of stuff in section 3.1. belongs to 2.2.

Response 3:  

Thank you for the observation. We would like to clarify that the release concentration of the contaminants is indeed specified in the manuscript. As stated in lines 207–208, “Each internal source releases perchloroethylene (PCE) at a concentration of 10 mg/L, serving as the parent compound in a sequential biodegradation chain.” Other relevant parameters, including degradation and sorption coefficients, are summarized in Table 2.

In addition, in the Scenario with boundary injection (Section 3.2, Effect of sorption rate on dissolved-phase concentration), we assumed an inlet boundary source with a constant concentration of 1 mg/L for all species (PCE, TCE, DCE, VC, ETH), as mentioned in lines 246–247.

We acknowledge the proposed restructuring of the manuscript. However, we have decided to retain the current structure, where the model verification is presented in the Results section. This follows the convention adopted in many previous studies on analytical and semi-analytical models, where verification is either placed in the Results and Discussion section or presented as a standalone section, separate from the methodology (Sudicky et al., 2013; Chen et al., 2016; Ding et al., 2021). This is because verification typically involves not only solving benchmark cases, but also analyzing and discussing the model’s behavior and agreement with reference solutions.

Comments 4: Validation of the model with real-world data.

Response 4:

Thank you for your comment. We would like to clarify that this study focuses on developing and verifying a semi-analytical model for multispecies transport with multiple internal sources under rate-limited sorption conditions. As such, we have not conducted validation using real-world field data, primarily due to the lack of sufficiently detailed and time-resolved datasets that include source geometry, release history, and rate-limited sorption. We consider model validation using site-specific field data an important next step and plan to address this in future work.

Comments 5: Sensitivity analysis as well as uncertainty analysis.

Response 5:

Thank you for your valuable comment. We acknowledge the importance of conducting sensitivity and uncertainty analysis, especially for models involving multiple parameters such as degradation rates, sorption coefficients, and dispersion values.

However, the focus of this study is on the development and verification of a new semi-analytical solution method. A full-scale sensitivity or uncertainty analysis would require additional assumptions about parameter distributions and ranges, which would go beyond the current scope. To keep the manuscript focused and concise, we have not included such analyses here.

Comments 6: Scalability and Computational Efficiency.

Response 6:

Thank you for your comment. The proposed semi-analytical model is designed to be computationally efficient, as it does not require spatial or temporal discretization like conventional numerical methods. Concentration values at any point in space and time can be calculated directly, without solving large systems of equations.

Regarding scalability, the model can be readily extended to simulate multiple contaminant species and multiple internal sources. Interactions between species are captured through a sequential degradation framework. These characteristics make the model well-suited for evaluating complex contamination scenarios with minimal computational cost.

Comments 7: Application of the method with other contaminants.

Response 7:

Thank you for your comment. In this study, we applied the model to simulate the biodegradation chain of chlorinated solvents (PCE → TCE → DCE → VC → ETH) as a representative case. However, the proposed model is general in structure and can be applied to other contaminants that undergo first-order decay or sequential transformation, including petroleum hydrocarbons, radionuclides, or certain reactive metals.

References

Moridis, G.J. and Reddell, D.L., The Laplace transform finite difference method for simulation of flow through porous media, Water Resour. Res. 1991, 27 (8), 1873–1884.

Chen, J.S., Chen, C.S., Gau, H.S., Liu, C.W. A two-well method to evaluate transverse dispersivity for tracer tests in a ra-dially convergent flow field. J. Hydrol. 1999, 223(3–4), 175–197.

Sudicky, E. A., Hwang, H. T., Illman, W. A., Wu, Y. S., Kool, J. B., & Huyakorn, P. A semi-analytical solution for simulating contaminant transport subject to chain-decay reactions. J. Contam. Hydrol. 2013, 144(1), 20-45.

Chen, K., Zhan, H., & Zhou, R. Subsurface solute transport with one-, two-, and three-dimensional arbitrary shape sources. J. Contam. Hydrol. 2016, 190, 44-57.

Ding, X. H., Feng, S. J., & Zheng, Q. T. A two-dimensional analytical model for contaminant transport in a finite domain subjected to multiple arbitrary time-dependent point injection sources. J. Hydrol. 2021,  597, 126318.

Round 2

Reviewer 2 Report

Comments and Suggestions for Authors

authors addressed my comments well